# SCRaMbLE: A Study of Its Robustness and Challenges through Enhancement of Hygromycin B Resistance in a Semi-Synthetic Yeast

**DOI:** 10.3390/bioengineering8030042

**Published:** 2021-03-23

**Authors:** Jun Yang Ong, Reem Swidah, Marco Monti, Daniel Schindler, Junbiao Dai, Yizhi Cai

**Affiliations:** 1Manchester Institute of Biotechnology (MIB), The University of Manchester, 131 Princess Street, Manchester M1 7DN, UK; j.y.ong@uu.nl (J.Y.O.); reem.swidah@manchester.ac.uk (R.S.); marco.monti@postgrad.manchester.ac.uk (M.M.); Daniel.Schindler@mpi-marburg.mpg.de (D.S.); 2Department of Chemical Biology and Drug Discovery, Utrecht Institute for Pharmaceutical Sciences, and Bijvoet Center for Biomolecular Research, Utrecht University, Universiteitsweg 99, 3584 CG Utrecht, The Netherlands; 3Max-Planck Institute for Terrestrial Microbiology, Karl-von-Frisch-Straße 10, 35043 Marburg, Germany; 4CAS Key Laboratory of Quantitative Engineering Biology, Shenzhen Institutes of Advanced Technology, Chinese Academy of Sciences, Shenzhen 518055, China; junbiao.dai@siat.ac.cn; 5Guangdong Provincial Key Laboratory of Synthetic Genomics, Shenzhen Institutes of Advanced Technology, Chinese Academy of Sciences, Shenzhen 518055, China; 6Shenzhen Key Laboratory of Synthetic Genomics, Shenzhen Institutes of Advanced Technology, Chinese Academy of Sciences, Shenzhen 518055, China

**Keywords:** *Saccharomyces cerevisiae*, synthetic yeast, SCRaMbLE, hygromycin B, accelerated evolution, Sc2.0

## Abstract

Recent advances in synthetic genomics launched the ambitious goal of generating the first synthetic designer eukaryote, based on the model organism *Saccharomyces cerevisiae* (Sc2.0). Excitingly, the Sc2.0 project is now nearing its completion and SCRaMbLE, an accelerated evolution tool implemented by the integration of symmetrical loxP sites (loxPSym) downstream of almost every non-essential gene, is arguably the most applicable synthetic genome-wide alteration to date. The SCRaMbLE system offers the capability to perform rapid genome diversification, providing huge potential for targeted strain improvement. Here we describe how SCRaMbLE can evolve a semi-synthetic yeast strain housing the synthetic chromosome II (synII) to generate hygromycin B resistant genotypes. Exploiting long-read nanopore sequencing, we show that all structural variations are due to recombination between loxP sites, with no off-target effects. We also highlight a phenomenon imposed on SCRaMbLE termed “essential raft”, where a fragment flanked by a pair of loxPSym sites can move within the genome but cannot be removed due to essentiality restrictions. Despite this, SCRaMbLE was able to explore the genomic space and produce alternative structural compositions that resulted in an increased hygromycin B resistance in the synII strain. We show that among the rearrangements generated via SCRaMbLE, deletions of YBR219C and YBR220C contribute to hygromycin B resistance phenotypes. However, the hygromycin B resistance provided by SCRaMbLEd genomes showed significant improvement when compared to corresponding single deletions, demonstrating the importance of the complex structural variations generated by SCRaMbLE to improve hygromycin B resistance. We anticipate that SCRaMbLE and its successors will be an invaluable tool to predict and evaluate the emergence of antibiotic resistance in yeast.

## 1. Introduction

In the many years of scientific studies to understand the fundamentals of life, molecular biology has seen many breakthroughs, however much mystery still remains. Is our current understanding of biological systems sufficient to create and improve on nature? In 2008, we saw the first complete synthesis of the blueprint of life—the 582 kb genome of the bacteria *Mycoplasma genitalium* [1]. Now, the synthesis of an entire eukaryotic genome is on the horizon: a designer version of the 12 Mb genome of *Saccharomyces cerevisiae*, also known as baker’s yeast. This time around, the goal has been not only to synthesise the complete genome, but also to incorporate novel design features [2].

Several rational design principles form the basis of the Sc2.0 genome [2]. One of the most prominent changes is the implementation of SCRaMbLE. The *synthetic chromosome recombination and modification by LoxP-mediated evolution* (SCRaMbLE) system is initiated by the introduction of symmetrical loxP sites (loxPsym) 3 bp downstream of most non-essential genes. This allows massive genomic rearrangements upon induction of the Cre recombinase, creating a pool of highly diverse genotypes. The application of SCRaMbLE has proven its power to generate genomic diversity through condition-specific strain evolutions [3,4]. These have led to enhanced growth on an alternative carbon source (xylose) [5], improved production from heterologous pathways for compounds such as violacein [5,6], penicillin, carotenoids [6,7] and betulinic acid [8], and increased resistance to alkaline conditions, high temperature, ethanol and acetic acid [9,10].

Given its potential, we anticipate that SCRaMbLE could be a powerful tool to combat a current threat to the health of humankind—the emergence of antibiotic resistance in pathogens. Despite the gravity of our current situation, the discovery of new antibiotics is a painfully slow process. We must therefore anticipate and prepare for antibiotic resistance early if we hope to stop it from happening, or to have the solution ready for when it occurs. Selection for resistance happens naturally in microbes over generations. However, accelerated evolution offered by technologies such as SCRaMbLE allows us to expedite the process and to anticipate possible future emergence of antibiotic resistance in pathogens. Among all pathogenic infections, fungal infections kill more than 1.5 million and affect over a billion of people every year [11]. Taking the generally recognised as safe (GRAS) *S. cerevisiae* yeast as an example of fungi and a widely used laboratory antibiotic hygromycin B, we use SCRaMbLE to evolve and deconvolute novel genotypes leading to hygromycin B resistance.

The choice of antibiotic in this study, hygromycin B, is an aminoglycoside antibiotic produced by *Streptomyces hygroscopicus* that was first isolated in 1958 [12]. Active against both prokaryotes and eukaryotes, hygromycin B antibiotic activity is attributed to the inhibition of protein synthesis [13] as it binds to RNA helix 44 (h44) of the small (30S) ribosomal subunit, next to the aminoacyl-tRNA binding (A) site [14,15,16]. Hygromycin B weakly stabilises the A site, but its potency results primarily from its inhibition of mRNA and tRNA translocation by blocking the tRNA path between the A and peptidyl-tRNA (P) sites [13,17].

Some bacteria have developed resistance mechanisms towards hygromycin B. For example, in *S. hygroscopicus* itself, the *hyg* gene codes for hygromycin B 7”-O-kinase (HYG; EC 2.7.1.119), which phosphorylates hygromycin B at the 7”-hydroxyl group of the destomic acid ring [18,19,20]. In *E. coli*, *hph* codes for hygromycin B 4-O-kinase (HPH; EC 2.7.1.163), which phosphorylates hygromycin B at the 4-hydroxyl group of the 2-DOS ring [21]. *hyg* and *hph* have no homology [19], and they phosphorylate hygromycin B at different sites. The phosphorylated hygromycin B products are inactive and do not affect protein synthesis [19]. Alternatively, different point mutations in ribosomal RNA genes have been identified in prokaryotic and eukaryotic cells that provide resistance to hygromycin B, i.e., U1406C, C1496U and U1498C in 16S rRNA in *Mycobacterium smegmatis* [22], mutation of G1491 or C1409 in 16S rRNA in *E. coli* [23], and U1711C in 17S rRNA in *Tetrahymena thermophila* [24]. In these cases, resistance to hygromycin B may have arisen from subtle changes in secondary structure of rRNA leading to weakened interaction with hygromycin B.

The *hph* resistance gene from *E. coli* has been used as a selectable marker in eukaryotic cells including yeast [25,26]. Very few other genes linked to hygromycin B resistance have been described in yeast [27,28,29]. In this study we employed SCRaMbLE to accelerate the evolution of a semi-synthetic yeast strain (synII), which harbours the 770 kb synthetic chromosome II [30], in the presence of hygromycin B. This allowed us to introduce stochastic genome rearrangements on chromosome II, including deletion, inversion, duplication and translocation, to identify novel gene rearrangements which may improve yeast’s resistance towards the antibiotic hygromycin B. Furthermore, we aim to gain insights into the capacity of the existing SCRaMbLE system to drive genotypic and phenotypic evolution by inspecting its capability to generate gene deletions which have been previously reported to increase hygromycin B resistance in yeast.

## 2. Results and Discussion

### 2.1. Generating a Hygromycin B Resistant Strain Using SCRaMbLE

To investigate whether using SCRaMbLE with the synII strain can improve its hygromycin B resistance, the parental synII strain (YCy1188) and the wild type strain (BY4742) were initially benchmarked for their resistance towards hygromycin B. Hygromycin B is usually used at the final concentration of 200 μg/mL for selection in yeast with the *hph* selectable marker [25,26]. A hygromycin B serial dilution spot assay was carried out using four biological replicates for each strain. For both BY4742 and synII, the lethal concentration was identified to be 150 μg/mL of hygromycin B after 2 days (Appendix A). Interestingly, synII was observed to show a slightly higher resistance to hygromycin B than BY4742, as shown by their differential growth at 150 μg/mL after 6 days of incubation. Our hypothesis is that the apparent increased resistance is due to the deletion of tRNA genes in synII, which has been reported to cause an up-regulation of translational machineries in the synII strain [30]. The up-regulation may have mitigated the binding of hygromycin B to the ribosomes subunit and therefore lessened the inhibition of protein synthesis. Similar up-regulation of the translational machinery was also observed in a *Leishmania donovani* strain resistant to paromomycin, another aminoglycoside antibiotic [31].

The SCRaMbLE experiment workflow is depicted in Figure 1. The Cre recombinase expression plasmid, pSCW11-Cre-EBD [32] was transformed into the synII strain (YCy2918) [3]. SCRaMbLE was induced for 24 h at 30 °C with 1 μM ß-estradiol in 10 mL selective SCD-His liquid media in order to maintain the pSCW11-Cre-EBD vector. Subsequently, the cells and the respective control (YCy2917 and YCy2919) were back-diluted to 0.1 OD_600_ and plated onto the YPD plates containing 200 μg/mL of hygromycin B. This was followed by 3–6 days of incubation at 30 °C. Thirty-three hygromycin B resistant SCRaMbLEd strains which grew successfully at the concentration of 200 μg/mL of hygromycin B, were isolated and submitted for further characterisation. These strains were picked in the order of decreasing colony size as a tentative measure of their resistance. They were termed HYG2.1 to HYG2.33.

The first ten of the 33 SCRaMbLEd strains, HYG2.1 to HYG2.10, were characterised in depth for their hygromycin B resistance via a spot test (Figure 2). HYG2.1 (YCy4021) showed the highest resistance level among the ten SCRaMbLEd strains tested, being able to grow at 250 μg/mL hygromycin B, outperforming the parental strain by 40%.

### 2.2. PCRTag Analysis for SCRaMbLEd Hygromycin B Resistant Strains

Following SCRaMbLE, we carried out quick genotyping to identify gene deletions in five resistant SCRaMbLEd strains, namely HYG2.1, HYG2.2, HYG2.3, HYG2.4 and HYG2.5 (YCy4021, YCy4022, YCy4023, YCy4024, YCy4025) (Appendix A) using PCRTag analysis.

PCRTag is a barcode system incorporated into Sc2.0 chromosomes by synonymous codon recoding of two ~20 bp regions, separated by around 500 bp, within each open reading frame (ORF) [2]. This allows us to distinguish between synthetic and wild type sequences via a simple PCR setup using primers for either the synthetic or the wild type allele. Furthermore, as applied here, PCRTag analysis allows us to detect the presence or absence of each synthetic ORF after SCRaMbLE.

Two gene deletions *IML3* (*YBR107C*) and *AIM3* (*YBR108W*) were detected in HYG2.1 (*cf.* DEL-1), while a deletion of the gene *GRX7* (*YBR014C*) was detected in HYG2.2 (*cf.* DEL-5). Multiple deletions were detected in HYG2.2 and HYG2.4 strains and summarized in Appendix A. All of these deletions were confirmed by nanopore sequencing (see Section 2.3). 

The PCRTag analysis is only able to indicate the presence or absence of a gene. PCRTag analysis cannot therefore identify inversions, duplications or translocations, which are other possible events resulting from SCRaMbLE. In this case, a more thorough analysis of the structural variations in the SCRaMbLEd strains is made possible by genome sequencing.

### 2.3. Nanopore Sequencing Analysis for Hygromycin B Resistant Strains

Short read sequencing has been shown to be inadequate for solving complex structural variations in SCRaMbLEd genomes [4]. However, third generation sequencing techniques, such as nanopore sequencing, provide ultralong reads that can be used to solve highly complex structural variations in SCRaMbLEd yeast [5]. We therefore decided to analyse the genomes of three strains with the highest hygromycin B resistance (HYG2.1, HYG2.2 and HYG2.4) via nanopore sequencing.

Our Nanopore sequencing result has a coverage from 7.15 to 13.23-fold with a mean read length of 16 kb and an N50 of 36 kb. The longest obtained read which mapped to the genome was 171 kb. An overview of the sequencing data is available in (Figure 3D).

Interestingly, the sequencing data indicated an additional copy of *TSC10* (*YBR265W*) in chromosome 8 (INS-0) in all SCRaMbLEd isolates as well as in the parental synII strain (YCy1188). This was found to be the result of an off-target integration of the wild type *TSC10-URA3* at *ARS810* in chromosome 8. The wild type *TSC10-URA3* integration was intended to replace the synthetic *YBR265W* gene in synII (YCy1189) as it was identified as the cause of a growth defect [30]. Our sequencing data showed that homologous replacement did not take place but wild type *TSC10* was integrated as an additional copy on chromosome 8 (Appendix A).

All other structural variations detected were between loxPsym sites, confirming that structural variations were caused by Cre recombinase. Appendix A highlights all detected structural variations within the sequenced strains, where each structural variation (INS = insertion, DEL = deletion, INV = inversion, DUP = duplication, INVDUP = inverted duplication) is accompanied by a number (0, 1, 2, etc., in the order mentioned) for cross-reference on the dot plot (Figure 3).

The strain HGY2.1 (YCy2934) contains a single deletion (DEL-1) of approximately 4.5 kb carrying both *IML3* (*YBR107C*) and *AIM3* (*YBR108W*) genes. Iml3 plays a role in kinetochore function and a null mutation leads to defects in the segregation of chromosomes and minichromosomes [33]. The Aim3 protein works together with Abp1 to inhibit barbed-end actin filament elongation [34]. Neither of these genes have been associated with increased resistance to hygromycin B in the WT background strain, or other aminoglycoside antibiotics through systematic mutation sets, and their functions indicate no obvious contribution to hygromycin B resistance. However, it is possible that deletions of *IML3* (*YBR107C*) and *AIM3* (YBR108W) in the synII strain could have a synergistic effect and associate somehow with the resistance phenotype in the synthetic background strain.

Strain HYG2.2 (YCy2935) and HYG2.4 (YCy2937) showed more complex variations, with in total eleven SCRaMbLE events between them. In HYG2.2 strain, we solved a complex structural variation consisting of a 79 kb duplication and inversion event (INVDUP-2), alongside an inversion of 34 kb in the same region (INV-3). DEL-4 is an additional 1 kb deletion which does not contain an ORF. Furthermore HYG2.2 contains a 1.3 kb deletion encoding *GRX7* (DEL-5) and a 14 kb duplication (DUP-6). Finally, a 7 kb sequence encoding for *BIT2*, *EFM2* and *HSM3* is inverted (INV-8).

Strain HYG2.4 (YCy2937) shows evidence of four SCRaMbLE events, including one 2 kb centromeric deletion containing *ARS208* and *YBL001C* (DEL-9). However, the area surrounding the centromere was duplicated but the centromere was deleted in one of the duplicated segments. Furthermore, we found a 13 kb inversion containing seven genes (*YBR050C*, *YBR051W*, *YBR052C*, *YBR053C*, *YBR054W*, *YBR055C*, *YBR056W*) (INV-10), a deletion of 14 kb (DEL-11) which was not detected by PCRTag analysis, and a 7 kb inversion containing four genes (INV-12).

The identified gene deletions presented an opportunity to match the variations in genotype with the observed resistance phenotype.

Accordingly, we proceeded to evaluate the impact of the identified gene deletions towards hygromycin B resistance in a WT background as the single knockout (KO) yeast collection [35] is available resources in the lab for further characterization study in the lab and synII strain was originated from BY4741. A total of 22 corresponding single deletion strains were selected from the single knockout (KO) yeast collection [35] for spot test analysis on YPD containing different concentrations of hygromycin B (Figure 4). Surprisingly, both *iml3Δ* and *aim3Δ* single KO strains exhibited a sensitive phenotype towards hygromycin B compared to BY4742, synII and HYG2.1. This suggests that the resistance phenotype of HYG2.1 generated during SCRaMbLE was not caused by either *IML3* or *AIM3* deletion alone. Interestingly, we observed small but noticeable improvements in hygromycin B resistance in single knockout strains of two uncharacterized genes *YBR219C*, *YBR220C*. In addition, slight hygromycin B resistance improvement was associated with *YBR084C-A* deletion. Effects of these gene deletions on hygromycin B resistance have not been reported before.

None of the single deletion strains were shown to be able to compete with the resistance level shown by the SCRaMbLEd strains harbouring complex, combinatorial rearrangements. This demonstrates the power of SCRaMbLE to explore complex structural variations beyond single deletions to rapidly produce desired phenotypes.

For this reason, we investigated whether combining the observed deletions in a double knockout strain could reproduce the observed hygB resistance phenotype. We decided to generate two double deletion mutants based on our single knockout strain analysis. We hypothesized that these deletions are the major contributing factors for the observed phenotype of HYG2.1 and HYG2.4. HYG2.1 contains the deletion of *YBR107C* and *YBR108W.* The individual deletions do not improve the phenotype (Figure 4). HYG2.4 accumulated multiple deletions but in the single knockout analysis only *YBR219C∆* and *YBR220C∆* slightly improve the hygB resistance phenotype (Figure 4). Therefore, the double deletion strains *YBR107C∆ YBR108W∆* and *YBR219C∆ YBR220C∆* strains were constructed in the BY4742 or BY4741 starting from single strains out of the knockout (KO) yeast collection [35]. The double deletion strain *YBR107C∆ YBR108W∆* indicates a slight improvement compared to the single knockout strains *YBR107C∆* and *YBR108W∆* on YPD supplemented with 100 µg/mL hygromycin B. However, the level of the hygB resistance improvement was less compared to HYG2.1. In contrast, the double knockout *YBR219C∆ YBR220C∆* strain has a severe growth defect even under standard growth conditions and does not improve the resistance phenotype towards hygB (Figure 5). It is more likely the resistance phenotype in the SCRaMbLEd strains is related to the complex rearrangements generated via SCRaMbLE which play a synergistic effect towards hygB resistance phenotype. Future transcriptome and proteome analysis may shed light into the complex situation causing the observed phenotype. Potentially, the effects might be accentuated by or particular to the synthetic yeast background.

### 2.4. Yeast Knockout Library Assessment of Hygromycin B Resistance

Whilst SCRaMbLE successfully produced strains with improved hygromycin B resistance, we sought to inspect the capability of SCRaMbLE to generate the optimal genomic variations for this trait. We assessed whether the variations generated by SCRaMbLE mentioned above could outcompete known hygromycin B resistant strains based on large-scale screening of the YKO collection. In chromosome 2, the deletion of *NCL1* (*YBL024W*) [28], *ECM8* (*YBR076W*) [29], *FES1* (*YBR101C*) [28] and *AGP2* (*YBR132C*) [27] were previously reported to cause a hygromycin B resistant phenotype. For the purpose of this study, two single knockout strains were selected as positive controls for hygromycin B resistance, *fes1Δ*::kanMX6 (YCy2983) and *agp2Δ*::kanMX6 (YCy2984) [35].

*fes1Δ* and *agp2Δ* were compared to the resistant SCRaMbLEd strains HYG2.1-HYG2.5 (Figure 6A). *fes1Δ* strain showed a noticeably higher resistance to hygromycin B compared to BY4742 and synII, in agreement with the previous report [28], as well as compared to SCRaMbLEd strains HYG2.1- HYG2.5. Unexpectedly, the *agp2Δ* strain showed high sensitivity to hygromycin B, in contrast to what was previously reported [27].

Fes1 is an Hsp70 nucleotide exchange factor, essential for the degradation of misfolded proteins by the ubiquitin-proteasome system. *fes1Δ* has been associated with an unusually strong and constitutive heat shock response at 25 °C [36,37]. *fes1Δ* has also been reported to cause bleomycin resistance in *S. cerevisiae* [38]. However, the mechanism via which *fes1Δ* causes resistance towards bleomycin or hygroymycin B is unclear.

We explored why *FES1* deletion was not detected in SCRaMbLEd strains despite the exceptional capacity of SCRaMbLE to generate other structural variations leading to higher hygromycin B resistance, albeit those less effective than *fes1* deletion itself. Scrutiny of the synII chromosome map revealed that during SCRaMbLE, *FES1* is linked to an essential gene. The loxPsym sites flanking *FES1* enclose three other genes: *SIF2*, *YMC2* and the essential *EXO84* (Figure 6B). This is due to the Sc2.0 design principle where loxPsym sites are only located 3 bp downstream of non-essential genes.

We termed this connection of multiple essential/non-essential genes in a single loxP flanked unit an “essential raft”, where the affected genes can move together but cannot be deleted from the synthetic genome. This highlights a limitation in the ability of the SCRaMbLE system.

Despite the drawbacks, we recognise this limitation as an opportunity for our research goal. The generation of a single deletion would already be possible with other techniques, while SCRaMbLE is better poised for more complex gene rearrangements. It was without the domination of *FES1* deletion in the SCRaMbLEd strains that allowed us to identify other structural variations that led to increased resistance and have not been reported before.

Despite the undesired effect associated with the *fes1* essential raft, we attempted to further increase hygromycin B resistance by carrying out additional rounds of SCRaMbLE on HYG2.1, the strain that demonstrated the one of the highest resistances after the first round of SCRaMbLE. However, no candidates with further improvement of hygromycin B resistance were obtained from two biological replicates. As a result, we reasoned that the first round of SCRaMbLE in this genotype background was already sufficient to maximise hygromycin B resistance in the context of a single synthetic chromosome II. This may have reached the local minimal in the genetic space, or alternatively the additional genomic rearrangements from further SCRaMbLE rounds may have resulted in reduced fitness in descendent strains, a phenomenon also described previously by Blount and coworkers [5].

## 3. Materials and Methods

### 3.1. Strains and Media

YCy2915 [BY4742 (*MATα his3Δ1 leu2Δ0 lys2Δ0 ura3Δ0*)] [39,40] and YCy1188 [synII (*MATa his3Δ1 leu2Δ0 LYS2 met15Δ0 ura3Δ0 synII::URA3*)] [30] were used as controls. The pSCW11-cre-EBD plasmid was introduced into SynII and used for the induction of SCRaMbLE. All SCRaMbLEd strains generated in this study are listed in Appendix A. Strains were grown on either standard yeast extract/peptone/dextrose media (YPD) for non auxotrophy-selective yeast growth, with hygromycin B added to the indicated concentration, or synthetic complete dextrose media (SCD) for auxotrophy-selective growth, both supplemented with 2% glucose [41]. Yeast media components were supplied by Fisher Scientific and Formedium (Norfolk, UK).

### 3.2. Yeast Transformations

All yeast transformations were performed using the lithium acetate method with a 20 min heat shock at 42 °C prior to plating on appropriate selective media [42].

### 3.3. Genomic and Plasmid DNA Isolation

Genomic DNA was isolated for PCRTag analysis using phenol-chloroform extraction [43]. Plasmids were isolated from bacterial hosts using the QIAprep spin Miniprep Kit (Qiagen) (Hilden, Germany).

### 3.4. Double Deletion Strains

Double deletion strains were constructed starting from a single deletion strain derived from KO yeast collections [44] either BY4742 or BY4741. The *URA3* deletion cassette was generated by PCR and transformed into the respective single deletion strain. The target gene was deleted using homologues recombination machinery. The confirmation PCR was performed to verify upstream and downstream chromosomal integration sites.

### 3.5. SCRaMbLE Workflow

A single colony of the semi-synthetic synII strain (YCy1188) bearing pSCW11-cre-EBD was inoculated in SCD-His and grown overnight at 30 °C. The overnight culture was used to inoculate 10 ml of SCD-His to an OD_600_ of 0.1. β-estradiol (Sigma Aldrich) (Munich, Germany).was added to a final concentration of 1 μM to induce Cre expression and SCRaMbLE in cells. Cultures were grown for 24 h, shaking at 30 °C. Cultures were back diluted to 0.1 OD_600_ and 50 μL aliquots were plated onto YPD plates containing different concentrations of hygromycin B (Thermo) (Dreieich, Germany) without selection for pSCW11-cre-EBD. Plates were incubated at 30 °C for 2–3 days in order to select for hygromycin B resistant strains.

### 3.6. Post-SCRaMbLE Selection for Hygromycin B Resistance

Following SCRaMbLE of synII (YCy1188), single colonies of resistant strains were selected for further characterisation based on the colony size and ability to grow at 200 μg/mL of hygromycin B. Then, a spot test assay was performed on YPD plates containing different concentrations of hygromycin B. For this, each strain was inoculated into 5 mL of YPD medium for 24 h, then re-inoculated to obtain a serial dilution starting with 0.1 OD_600_. 

### 3.7. PCRTag Analysis of HYG2.1 to HYG2.5 Strains

For each genomic DNA sample, the following master mix was prepared: 6.25 μL of DreamTaq Green PCR Master Mix (2X) (ThermoFisher Scientific), 2.5 μL of genomic DNA (20 ng/μL) and 2.25 μL of sterile distilled water. 11 μL of the master mix was aliquoted into each well of a 96-well PCR plate and 1.5 μL of pre-mixed forward and reverse PCRTag primers (10 mM) were added. The PCR thermal-cycler program was as follows: 94 °C/3 min, 30 cycles of (94 °C/30 s, 60 °C/30 s, 72 °C/30 s), and a final extension of 72 °C/7 min. PCR samples were loaded directly onto a 1% agarose gel for electrophoresis and bands were visualized using a ChemiDoc™ XRS+ System (Bio-Rad). The presence or the absence of the amplicon was assessed visually.

### 3.8. Nanopore Sequencing

High quality DNA extraction was performed according to a modified Qiagen Genomic-tip 100/g protocol with the Qiagen Genomic Buffer kit. DNA quality was assessed via gel-electrophoresis, NanoDrop™ 2000 Spectrophotometer and Qubit 4 Fluorometer using dsDNA BR reagents. Library preparation was performed using the SQK-LSK108 library kit with the Native Barcoding kits EXP-NDB104 and EXP-NDB113. Kits were used largely according to the manufacturers’ guidelines, however input DNA was not sheared and the starting DNA concentration was increased 5-fold to match the molarity expected in the protocol. This was based on our experience with PFGE analysis of similarly prepared genomic DNA (data not shown). Sequencing was performed on a MinION Mk1B device using FLO-MIN106D with R9.4.1 chemistry. DNA sequencing was performed for 48 hours using the software MinKNOW v19.05.0.

Base calling and demultiplexing was performed locally using Guppy software (v3.1.5). Data obtained were mapped against the reference genome of BY4741 chrII::synII using minimap2 (v2.17) [45] and NGMLR (v0.2.7) [46]. Rearrangements in the NGMLR mapping data were called using Sniffles (v1.0.11) [46] with a threshold of ≥10 reads confirming the rearrangement. *De novo* genome assembly of each strain was performed using Canu (v1.8) [47]. SCRaMbLE rearrangements in synII were evaluated and confirmed by comparing the mapping and variant calling data from Sniffles with the de novo assembly obtained by Canu [47].

## 4. Conclusions and Perspectives

Past studies have shown that SCRaMbLE is a powerful yet versatile tool that can be used to evolve the genotype of the synthetic strains to tolerate increased stressors, such as alkaline condition, high temperature and ethanol [9,10]. In this study, we show that the current SCRaMbLE system in Sc2.0 allowed us to generate gene rearrangements/structural variations in synthetic yeast containing synthetic yeast chromosome II that give rise to improved hygromycin B resistance. The function of SCRaMbLE is analogous to a combinatorial black-box. Therefore, improved strains can be generated without prior knowledge by simply shuffling genes in the synthetic chromosome(s). Desired phenotypes are subsequently selected from the enormous pool of genotypes produced. This stochastic process sheds new light on the possibilities of large-scale studies that were previously infeasible when using pools of randomly mutagenised yeasts, or via synthetic lethal array technology.

Despite our effort, the mechanism behind the higher hygromycin B resistance generated remains elusive. Gene deletions identified in the resistant SCRaMbLEd strains led us to assess the hygromycin B resistance of the corresponding single knockout strains. We were able to identify two uncharacterized genes, *YBR219C* and *YBR220C*, deletion of which led to slight, previously unreported improvements in the hygromycin B resistance level. However, the individual gene deletions could not reach the resistance level conferred by complex structural variations generated by SCRaMbLE. The nanopore sequencing technology was used to deconvolute the structural rearrangements of the SCRaMbLEd genomes. However, the nanopore sequencing coverage within this study is not sufficient to confidently detect individual SNPs, but we have in the past performed comprehensive SCRaMbLE genome sequencing with Illumina technology and found the frequency of SNPs in these genomes are on par with their parental un-SCRaMbLEd [4]. Additionally, we have included a parental unSCRaMbLEd strain (synII) in this experiment. After selection, no resistance phenotype of the unSCRamBLEd synII could be obtained. This suggests it is unlikely that SNPs play are the causative of the observed HygB resistance phenotype. Nevertheless, increasing nanopore sequencing coverage or performing short read sequencing would be implemented in future characterization of SCRaMbLEd synthetic yeast strains.

Future in-depth investigations such as transcriptome profiling of multiple different resistant strains may indicate common patterns that could potentially point to the origin of the resistance.

Interestingly while we were attempting to inspect the capability of SCRaMbLE, we observed that the deletion of *AGP2* rendered the strain sensitive to hygromycin B, instead of becoming more resistant as has been reported previously [27]. Conversely, *FES1* knockout elicited the expected improvement in hygromycin B resistance [28]. Furthermore, we also investigated the reason why *FES1* deletion was not observed via SCRaMbLE. *FES1* is coupled to the essential gene *EXO84* as part of an essential raft. Deletion of *FES1* would be possible if the neighbouring essential gene *EXO84* in the essential raft is moved or integrated into a different locus in the genome. Alternatively, inserting an additional loxP downstream to the *FES1* gene using CRISPR would bypass this problem. This will likely further improve the resistance phenotype of the synthetic yeast after SCRaMbLE.

Notwithstanding, facing this challenge could present a new opportunity to improve SCRaMbLE. Systematic single deletions are routinely done without the necessity of synthetic chromosomes or SCRaMbLE. The SCRaMbLE system is, however, ideally poised to allow the investigation of more complex variations that are impossible through systematic mutations. Here we showed how SCRaMbLE was able to detect other complex genome modifications that yeast can develop for hygromycin B resistance. This may also put forward an alternative strategy, where the known resistance genes are deliberately coupled to essential genes so that effects of other genes can be assessed. Whether by itself or with the help of customised modifications, SCRaMbLE can help to anticipate the emergence of antibiotic resistance.

Supplying essential genes on a separate chromosome could be an alternative strategy to increase the deletion power of SCRaMbLE and increase the plasticity of the synthetic genome to obtain more desirable phenotypes. This strategy has been demonstrated by a recent paper [40] and proposed in the highly anticipated Sc3.0 project. It is also worth mentioning that this study was performed using only one synthetic chromosome. By using a fully synthetic strain and/or the improved SCRaMbLE system in Sc3.0, we should be able to obtain even more complex SCRaMbLE results and variants with higher antibiotic resistance.

## Figures and Tables

**Figure 1 bioengineering-08-00042-f001:**
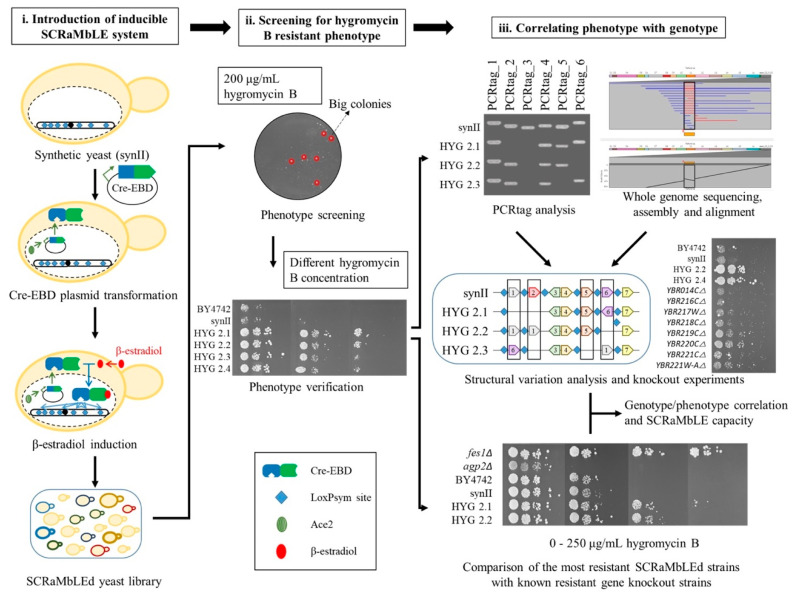
Schematics of a SCRaMbLE workflow to generate synthetic yeast with increased hygromycin B resistance. The workflow consists of three phases: (**i**) Introduction of the inducible SCRaMbLE system. The synthetic yeast is transformed with a pSCW11-Cre-EBD plasmid, expression of which is induced by estradiol to generate a library of SCRaMbLEd yeast strains; (**ii**) Screening for SCRaMbLEd yeast strains with hygromycin B resistant phenotype. SCRaMbLEd strains are first spread onto plates containing 200 μg/mL hygromycin B concentration to select for resistant strains, phenotypes of which are then verified by spot test analysis on different hygromycin B concentrations; (**iii**) Identifying links between the genotype and the resistance phenotype, and inspecting the capacity of SCRaMbLE. The resistant strains are subjected to PCRtag analysis and whole genome sequencing to identify any structural variation that may have contributed to the resistance phenotype. Individual knockout strains derived from the identified variations were tested to identify the contribution of each gene to the resistance phenotype. On the other hand, knockout strains with higher resistance based on literature are compared to those improved strains obtained through SCRaMbLE to see whether the best possible strain has been generated.

**Figure 2 bioengineering-08-00042-f002:**
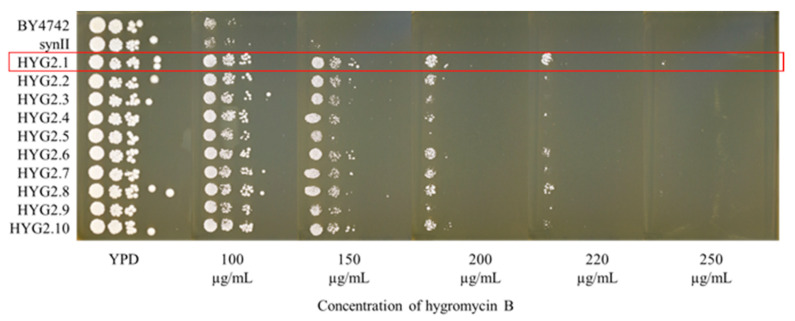
Spot test to assess hygromycin B resistance of the SCRaMbLEd strains HYG2.1-HYG2.10. SCRaMbLEd strains which grew at 200 μg/mL hygromycin B concentration were isolated and spotted on YPD plates containing different concentrations of hygromycin B. The plates were incubated at 30 °C and the photos were taken on day 5. SCRaMbLEd yeast strain HYG2.1 shows improved growth at higher hygromycin B concentrations compared to the wild type and the parental synII strains.

**Figure 3 bioengineering-08-00042-f003:**
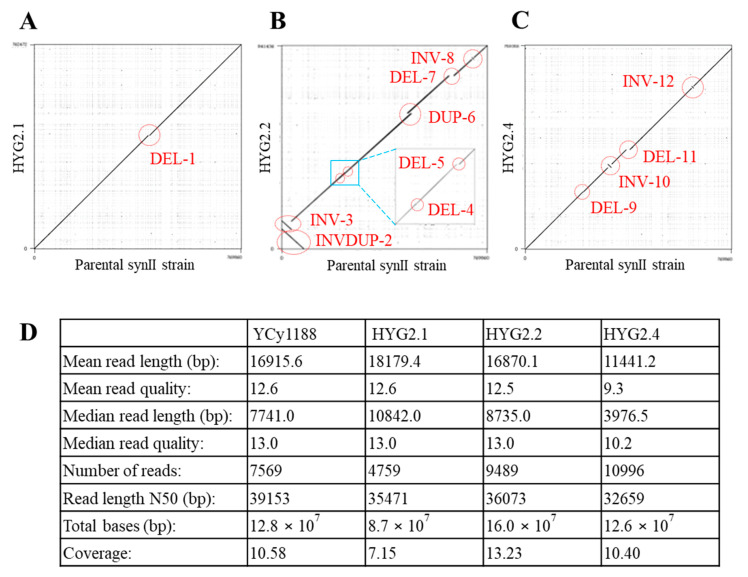
Dot plots illustrating the structural variations observed in the SCRaMbLEd strains HYG2.1 (**A**), HYG2.2 (**B**) and HYG2.4 (**C**) via nanopore sequencing. Each SCRaMbLEd strain (y-axis) is plotted against the parental synII (YCy1188) strain (x-axis) and each structural variation can be referenced to Appendix A. (**D**) Nanopore sequencing run data.

**Figure 4 bioengineering-08-00042-f004:**
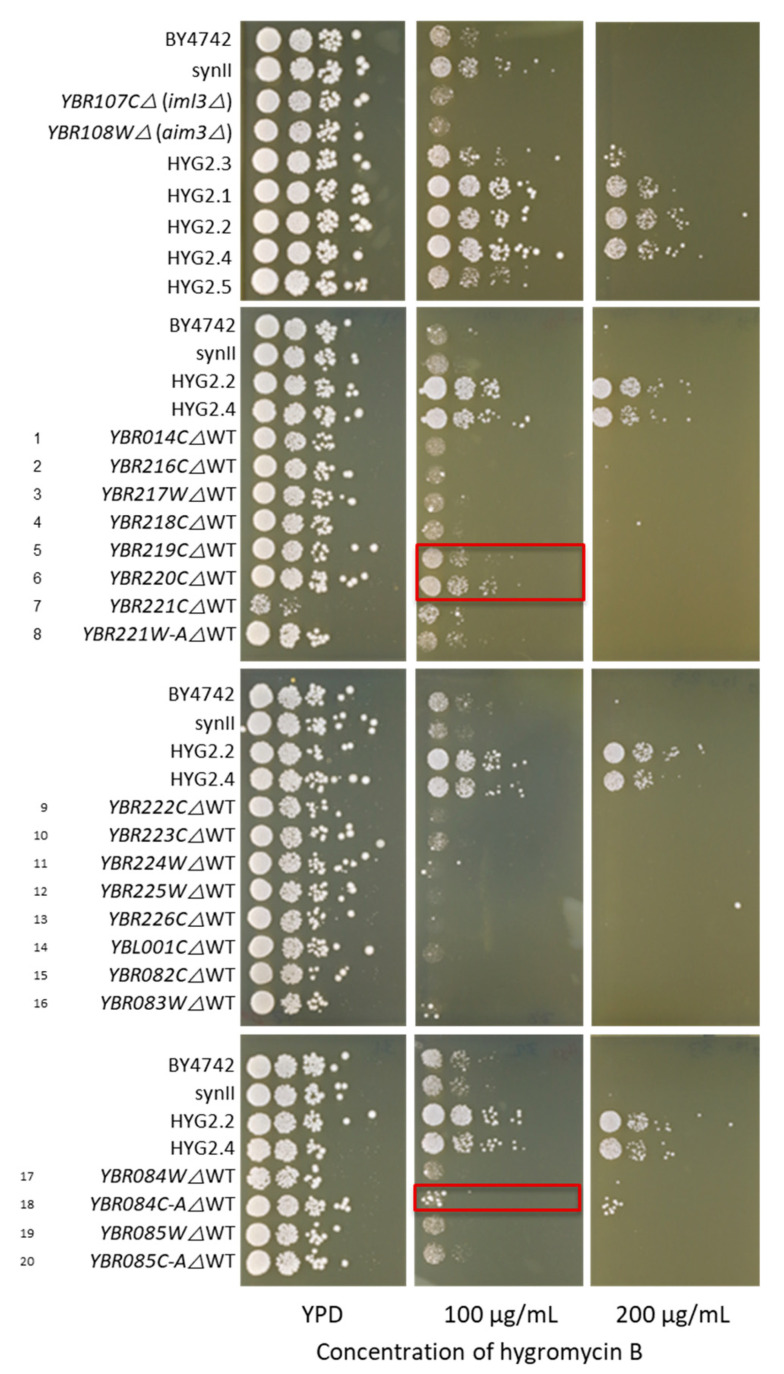
Phenotypic analysis showed that *YBR219C∆WT*, *YBR220C∆WT* and *YBR084C-A∆WT* contributed to hygromycin B resistance. Single knockout strains, BY4742, synII, SCRaMbLEd strains were spotted on YPD plates containing different concentrations of hygromycin B. Plates were incubated at 30 ℃ and the photos were taken on day 3. The increased of hygromycin B resistance is slightly increased when *YBR219C∆*WT, *YBR220C∆*WT and *YBR084C-A∆*WT were deleted individually from the WT strain. The hygromycin B resistance is significantly higher in the SCRaMbLEd strains HYG2.2 and HYG2.4 compared to single deletion strains.

**Figure 5 bioengineering-08-00042-f005:**
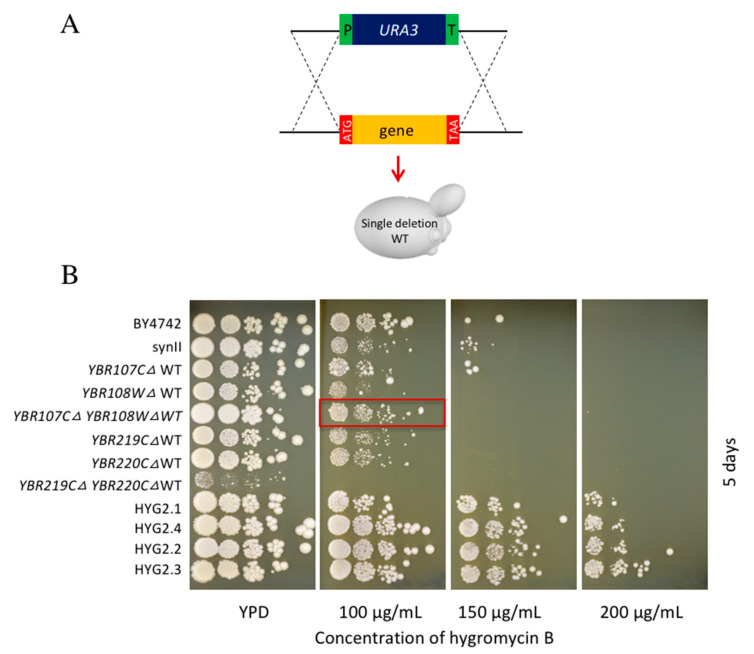
Phenotypic analysis showed that SCRaMbLEd strains are more resistant to hygromycin B compared to single/double deletions strains. (**A**) Visualization of the applied deletion strategy using the endogenous homologous recombination machinery. (**B**) Spot test. BY4742, synII, single knockout, double knockout and SCRaMbLEd strains were spotted on YPD plates containing different concentrations of hygromycin B. Plates were incubated at 30 ℃ and the photos were taken on day 5. The double deletion strain *YBR107C∆ YBR108W∆* WT grows slightly better compared to the single deletion *YBR107C∆* and *YBR108W∆* strains on the YPD plate +100 µg/mL hygromycin B. Interestingly, *YBR219C∆*. *YBR220C∆*WT strain has a severe growth defect on the normal growth conditions and it does not improve the resistance phenotype toward hygromycin B.

**Figure 6 bioengineering-08-00042-f006:**
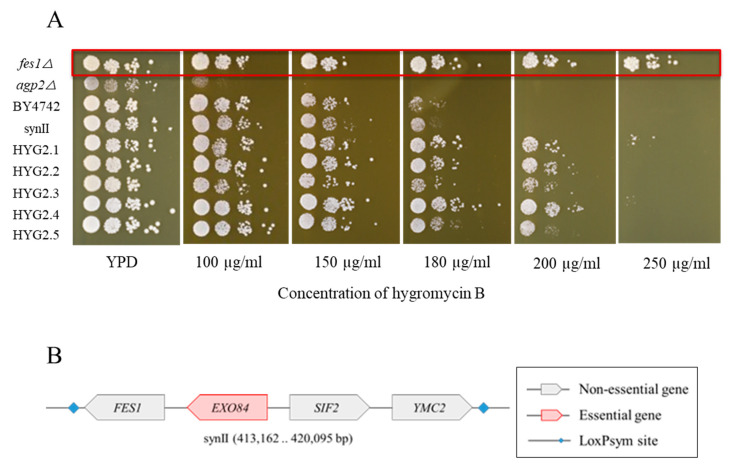
(**A**) Phenotypic analysis revealed that *fes1*Δ and *agp2*Δ had contrary effect on the hygromycin B resistance. Single knockout strains *fes1*Δ and *agp2*Δ, BY4742, synII, SCRaMbLEd strains HYG2.1-HYG2.5 were spotted on YPD plates containing different concentrations of hygromycin B. Plates were incubated at 30 ℃ and the photos were taken on day 3. *fes1*Δ increased the resistance towards hygromycin B [28], while *agp2*Δ led to sensitivity to hygromycin B in contrast to what was reported [27]. (**B**) The loxPsym sites flanking *FES1* enclose 3 other genes, *EXO84*, *SIF2* and *YMC2*, in which *EXO84* is an essential gene.

## Data Availability

All strains and vectors used in this study are available and can be obtained from the lab of Yizhi Cai (https://www.cailab.org, accessed on 17 March 2021). Any project details can also be provided upon request. All sequencing data are available on NCBI-Sequence Read Archive (SRA) under the BioProject ID PRJNA683650. The BioSample accessions are SAMN17034049, SAMN17034050, SAMN17034051 and SAMN17034052.

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
