# Peer review of "SCRaMbLE: A Study of Its Robustness and Challenges through Enhancement of Hygromycin B Resistance in a Semi-Synthetic Yeast"

_bioengineering, 2021, doi:10.3390/bioengineering8030042_

Round 1

Reviewer 1 Report

The ms is a revision and although most of the points raised by the reviewers have been dealt with, I could not find Table 1 anywhere and it is most certainly not Figure 3D, as pointed out by the Editor.

In the legend to Figure 4 there is text describing Table 1 and should be deleted.

Author Response

Responding to the reviewer comments

First of all, we would like to thank the reviewer for the effort to evaluate the submission. We are happy that we addressed most of the points raised and hope we can satisfy the reviewer with the following point by point response to the remaining minor points. Again, thank you very much for your time and effort for your time in handling our manuscript “SCRaMbLE: A study of its robustness and challenges through enhancement of hygromycin B resistance in a semi-synthetic yeast”.

Comments and Suggestions for Authors#

  1. The ms is a revision and although most of the points raised by the reviewers have been dealt with, I could not find Table 1 anywhere and it is most certainly not Figure 3D, as pointed out by the Editor.

It seems an accident happened and text was replaced by “Table 11”. The following text has been added: The workflow consists of three phases: i) Introduction of the inducible SCRaMbLE system. The synthetic yeast is transformed with a pSCW11-Cre-EBD plasmid. If the reviewer refers to table 1, which is mentioned in line 174, it can be found on the PowerPoint doc page 7.

  1. In the legend to Figure 4 there is text describing Table 1 and should be deleted.

The text regarding table 1 has been deleted.

Reviewer 2 Report

Regarding the issue of background SNPs in addition to structural rearrangements, since the authors did not perform short read sequencing to identify other possible types of genetic variation, I recommend adding a brief sentence or two in the discussion section to acknowledge that background mutations may be contributing to the resistance phenotypes observed.  Disentangling the effects of SNPs and structural rearrangements may certainly be out of the scope of the paper, but it is important to acknowledge this limitation of the study/future direction in the text.

Author Response

Responding to the reviewer comments

First of all, we would like to thank the reviewer for the effort to evaluate the submission. We are happy that we addressed most of the points raised and hope we can satisfy the reviewer with the following point by point response to the remaining minor points. Again thank you very much for your time and effort for your time in handling our manuscript “SCRaMbLE: A study of its robustness and challenges through enhancement of hygromycin B resistance in a semi-synthetic yeast”.

Comments and Suggestions for Authors

  1. Regarding the issue of background SNPs in addition to structural rearrangements, since the authors did not perform short read sequencing to identify other possible types of genetic variation, I recommend adding a brief sentence or two in the discussion section to acknowledge that background mutations may be contributing to the resistance phenotypes observed.  Disentangling the effects of SNPs and structural rearrangements may certainly be out of the scope of the paper, but it is important to acknowledge this limitation of the study/future direction in the text.

The reviewer is right and we added the following section from line 383-391.

The nanopore sequencing technology was used to deconvolute the structural rearrangements of the SCRaMbLEd genomes. However, the nanopore sequencing coverage within this study is not sufficient to confidently detect individual SNPs, but we have in the past performed comprehensive SCRaMbLE genome sequencing with Illumina technology and found the frequency of SNPs in these genomes are on par with their parental un-SCRaMbLEd12. Additionally, we have included a parental unSCRaMbLEd strain (synII) in this experiment. After selection, no resistance phenotype of the unSCRamBLEd synII could be obtained. This suggests it is unlikely that SNPs play are the causative of the observed HygB resistance phenotype. Nevertheless, increasing nanopore sequencing coverage or performing short read sequencing would be implemented in future characterisation of SCRaMbLEd synthetic yeast strains.

This manuscript is a resubmission of an earlier submission. The following is a list of the peer review reports and author responses from that submission.

Round 1

Reviewer 1 Report

This report by Ong, et al, describes the use of the Loxp sites in the SynII yeast strain to generate complex rearrangements in chromosome 2 and select for variants that are resistant to hygromycin B. The authors are successful in this endeavor, generating strains with several-fold increases to hygB resistance.  The authors then use nanopore sequencing to resolve the rearrangements present in the resistant strains, and test the impact of single knockouts of the genes that were deleted in the recovered strains.  Interestingly, a known deletion conferring HygB resistance (fes1) was not observed due to its linkage to essential genes.  This allowed novel genetic changes conferring hygB resistance to be observed.  I have several recommendations to make before I can recommend publication.

Major comments:

  1. The authors do not report whether there are other mutations (particularly SNPs) present in other chromosomes in the hygB-resistant strains. These could be very important to the resistance phenotypes observed, and it is unclear whether the coverage of nanopore reads they obtained are sufficient to detect SNPs, if they exist.  I recommend providing evidence that the nanopore coverage the authors obtained is sufficient to detect SNPs. If it is not, then follow-on experiments with short-read sequencing should be performed to detect SNPs and analyze their contributions to HygB resistance.
  2. The authors claim that SCRAMBLE can be used to understand the determinants of antibiotic resistance. However, none of the single knockouts tested approach the resistance level of the SCRAMBLEd strains, making it unclear how this resistance occurs.  The authors posit that perhaps the knockouts are synergistic, or that other rearrangements (e.g. duplications or inversions) are responsible. I recommend performing some follow-on assays (e.g. double knockouts or overexpressions) to further elucidate the contributions of the changes the authors observed.

Minor comments

  1. The first paragraph in the introduction is a bit grandiose. I recommend toning down the language.
  2. On page 4, there is a reference to Table 11 which I can’t find. Perhaps this is a typo?
  3. I recommend clarifying figure 4 to note from which strains the individual knockouts were found.
  4. A theoretical consideration of library sizes would be helpful. How many possible SCRAMBLE variants are there?  How many strains were plated?  How many recombinase events are there in each strain?  What is the coverage of the library?  This would help determine whether the recovered variants are “optimal” in the SCRAMBLE context.

Reviewer 2 Report

The ms describes the application of the SCRaMbLE system for the identification of gene re-arrangements in yeast which resulted in increased resistance to hygromycin B. Thirty-three  SCRaMbLEd strains displaying hygromycin B resistance were identified in the parental synII YCy188 strain by PCRTag followed by nanopore sequencing. The first ten strains were further characterised by spotting assays on media containing known concentrations of hygromycin B and deletions of the ORFs, YBR219C and YBR220C, were identified as conferring hygromycin B resistance, albeit to varying degrees.

This novel preliminary finding underlines the importance of complex chromosomal re-arrangements as a basis for hygromycin B resistance. Surprisingly, no deletion of FES1 (YBR101C) was detected in the SCRaMbLEd strains. The reason for this was that FES1 was not flanked by two loxPsym sites but was included in an approximately 7 kb region which also carried the essential EXO84 gene, meaning that this 7 kb region was an ‘essential raft’ and confirms the usefulness of  the ‘SCRaMbLing’ tool.

Nanopore sequencing also revealed that, unexpectedly, TSC10 (YBR265w) was located on chromosome VIII rather than on chromosome II. Further investigation indicated that this was also the case in the original semi-synthetic synII strain.

Minor issues:

On p4, the legend on Figure 1 suddenly contains Table 11, maybe taken from a PhD thesis??

In Figure 3(B) the labelling of INVDUP-2 should be consistent with that of Table 1.